# Identification of Four Mouse FcRn Splice Variants and FcRn-Specific Vesicles

**DOI:** 10.3390/cells13070594

**Published:** 2024-03-29

**Authors:** George Haddad, Judith Blaine

**Affiliations:** Division of Renal Disease and Hypertension, Department of Medicine, School of Medicine, University of Colorado, Aurora, CO 80045, USA; george.haddad@cuanschutz.edu

**Keywords:** neonatal Fc receptor, alternatively spliced variants, vesicular transport, immunoglobulin, albumin, immune complexes, cellular recycling, podocytes

## Abstract

Research into the neonatal Fc receptor (FcRn) has increased dramatically ever since Simister and Mostov first purified a rat version of the receptor. Over the years, FcRn has been shown to function not only as a receptor that transfers immunity from mother to fetus but also performs an array of different functions that include transport and recycling of immunoglobulins and albumin in the adult. Due to its important cellular roles, several clinical trials have been designed to either inhibit/enhance FcRn function or develop of non-invasive therapeutic delivery system such as fusion of drugs to IgG Fc or albumin to enhance delivery inside the cells. Here, we report the accidental identification of several FcRn alternatively spliced variants in both mouse and human cells. The four new mouse splice variants are capable of binding immunoglobulins’ Fc and Fab portions. In addition, we have identified FcRn-specific vesicles in which immunoglobulins and albumin can be stored and that are involved in the endosomal–lysosomal system. The complexity of FcRn functions offers significant potential to design and develop novel and targeted therapeutics.

## 1. Introduction

Almost 35 years ago, Simister and Mostov purified the first neonatal FcRn from rat intestine, which validated Brambell’s hypothesis from some 20 years prior [1]. Since that time, the functions of FcRn have been extended to not only transporting immunity from mother to fetus across the placenta [2] but to a multitude of other functions. As such, FcRn is best known for recycling monomeric IgG and albumin and thus increasing their half-life [3,4,5]. In addition, FcRn has been shown to play essential roles in phagocytosis, antigen presentation, immune complex degradation, and binding and protecting fibrinogen from intracellular destruction [6,7,8,9].

FcRn belongs to the Fcγ receptor family; however, structurally it is more closely related to class I major histocompatibility complex and forms a heterodimer complex with β2-microglobulin, which is important for proper FcRn folding and overall functioning [10,11]. FcRn is expressed predominantly in the cytoplasm and found in early (Rab5+ vesicles) and recycling (Rab11a+ vesicles) endosomes [12,13].

The role of FcRn in recycling and prolonging the half-life of IgG is essential in immunity against pathogenic foreign invaders. However, this very function can exacerbate the extent of tissue injury by protecting deleterious autoantibodies from lysosomal degradation. FcRn binding and recycling of IgG and albumin have been exploited to develop therapeutic strategies for enhanced drug delivery, such as Fc or albumin conjugated with viral antigens for efficient delivery of vaccines or to block IgG recycling to improve treatments of autoimmune diseases [11,14,15,16].

Indeed, a significant number of clinical trials are currently underway for either optimizing FcRn–IgG interaction to reduce drug dosage and enhance treatment effectiveness or to block FcRn–autoantibody interactions. The various clinical trials are reviewed in [17,18,19].

To facilitate investigation of FcRn trafficking of IgG and immune complexes, we sought to overexpress FcRn by cloning the receptor into a lentiviral expression vector. In this study, we report the accidental identification of four alternatively spliced FcRn variants in mouse and the initial characterization of the newly identified receptors. The original FcRn has been designated FcRn1 and the new four variants have been denoted FcRn2, FcRn3, FcRn4, and FcRn5. The discovery of the FcRn splice variants indicate the complexity of FcRn’s role in IgG and/or albumin recycling and vesicle trafficking. In addition, we have identified several alternatively spliced variants in human podocytes, suggesting perhaps a conserved mechanism of cargo sorting/recycling in mammalian cells. We also identified that FcRn is expressed in/on vesicles independently of early, late, and recycling vesicles, or lysosomes. We show that depending on the nature of the internalized cargo, FcRn vesicles interact with the other vesicles of the endosomal–lysosomal system according to the type of endocytosed materials. This interaction leads to either cargo recycling, lysosomal degradation, or cargo ejection outside the cell.

## 2. Materials and Methods

### 2.1. Reagents

Anti-FLAG tag antibody (Novus Biological, Centennial, CO, USA, cat# NBP1-06712SS, lot# C-7), LAMP1 (abcam, Boston MA, USA cat#ab208943, lot# GR3213103-23), Rab 5 (Cell Signaling, Danvers, MA, USA, cat#3547 (C8B1), lot# 7), Rab7 (Cell Signaling, cat#9367 (D95F2), lot# 3), Rab11 (Cell Signaling, cat#5589 (D4F5), lot# 4), mouse FcRn antibody (Invitrogen, ThermoFisher, Waltham, MA, USA cat# PA5-47871, lot# YA3813873), hexokinase 1 (abcam, cat# ab150423, lot# 1022187-1), 4-hydroxy-3-nitrophenylacetyl (NP)-ovalbumin (Biosearch Technologies, Radnor, PA, USA, cat# N-5051), anti NP-ova IgG1 antibody (clone N1G9) a kind gift from Dr. Raul Torres (University of Colorado), DreamTaq hot start master mix (ThermoFisher, Waltham, MA, USA, cat# K9011, lot# 01331773), AccuPrime *Pfx* DNA polymerases (ThermoFisher, cat# 12344-024, lot# 2588196), and mouse serum albumin (SigmaAldrich, St. Louis, MO, USA cat# A3139-5 mg).

### 2.2. Cell Culture

Mouse wild-type and FcRn-knockout podocyte cell lines derived from C57BL/6J and B6-129 × 1-*Fcgrt^tm1Dcr^*/DcrJ FcRn knockout mice, respectively, were used in this study. The podocyte cell lines were immortalized according to the protocol described in [20]. The cells were from previously frozen stocks and no mice or any animals were used in this study. The podocyte cell lines were cultured in RPMI1640 supplemented with 10% FBS and penicillin (100 U/mL)–streptomycin (100 μg/mL).

### 2.3. RT-PCR

RNA was extracted using the TRIzol (Invitrogen, Waltham, MA, USA) method and quantified using NanoDrop 2000 (ThermoScientific). One microgram of total RNA was reverse-transcribed into cDNA using LunaScript (NewEngland Biolabs, Ipswich, MA, USA, Cat# E3010L). The primers were ordered from Integrated DNA technologies IDT, San Diego, CA, USA, and the DreamTaq^TM^ Hot Start PCR master mix was purchased from ThermoFisher (Cat# K9011). A BioRad (Hercules, CA, USA) T100 thermocycler was used to amplify the genes of interest.

### 2.4. FcRn Cloning

Mouse FcRn specific cloning primers, sense: 5′ GGTACC CCACC ATG GGG ATG CCA CTG CCC T 3′, antisense: 5′ GCGGCCGC TCA *CTT GTC ATC TTT GTA GTC* GGA AGT GGC TGG AAA GGC ATT TGC A 3′ (italic letters indicate FLAG-tag sequence), human FcRn cloning primers, sense: 5′ GGTACC CCACC ATG GGG GTC CCG CGG 3′, antisense: 5′ GCGGCCGC TCA GGC GGT GGC TGG AAT CAC ATT TA 3′, were designed to amplify the full sequence using RT-PCR. The restriction enzyme sites for KpnI and NotI were included in the primers to facilitate directional cloning into the VVPW plasmid. The PCR product(s) was first cloned into topo-TA vector (ThermoFisher, Cat # K4575J10) and the resulted clones were validated by sequencing (Quintara Biosciences, Hayward, CA, USA).

### 2.5. Lentivirus Production

The FcRn clones were inserted into the lentivirus plasmid VVPW (kind gift from Gabriele Luca Gusella, Icahn School of Medicine Mount Sinai Hospital, New York City, NY, USA) using KpnI and NotI restriction sites and the clones were validated by sequencing. The VVPW plasmid along with the packaging plasmid psPAX2 (Addgene, Watertown, MA, USA) and the envelope plasmid pCMV-VSV-G (Addgene, Watertown, MA, USA) were combined with FUGENE HD transfection reagent at the 3:2:1 ratio and added to ~70% confluent 293T cells. The cell supernatants were collected after 48 and 72 h posttransfection and validated for FcRn splice variant overexpression by qPCR using RNA isolated from cells infected with 20% lentivirus supernatants. The mouse podocytes were infected with FcRn splice variant lentivirus particles in addition to 8 μg/mL polybrene. The cells were used in experiments after 72 h of infection.

### 2.6. Purification of FcRn Splice Variant Proteins

FcRn variants were purified from mouse podocytes infected with FcRn splice variant lentiviruses using a Dynabeads Protein G Immunoprecipitation kit (Invitrogen) as per the manufacturer’s instructions. Briefly, 10 μg of anti-FLAG tag antibody (Bethyl Laboratories, Waltham, MA, USA) was combined with 50 μL of the magnetic beads in binding buffer and incubated for 10 min at room temperature with rotation. Podocyte cell lysates (prepared with RIPA buffer supplemented with protease inhibitor cocktail) were mixed with magnetic beads and incubated for 2 h at room temperature with rotation. The beads were washed 3 times with washing buffer, the recombinant proteins were eluted using the non-denaturing method, and the pH was adjusted by adding 1 M Tris, pH 7.5.

### 2.7. Western and Far Western Blot

The cell lysate concentration was estimated using Pierce BCA protein assay and 10 μg was resolved in 10% SDS-PAGE gel. The proteins were transferred onto a PVDF membrane and blocked with 5% skim milk in PBST. The primary antibodies were added at 1:1000 dilution prepared in blocking buffer and incubated overnight at 4 °C. The membranes were washed 3 times with PBST for 30 min and anti-species secondary antibodies were added at 1:10,000 dilution for 1 h at RT. The membranes were washed again with PBS for 30 min and developed using Immobilon Forte (MilliporeSigma, Burlington, MA, USA) Western Blot Substrate and exposed to an X-ray film.

Fifty micrograms each of mouse anti NP-OVA IgG and mouse serum albumin were resolved on an SDS-PAGE gel under native and/or reducing and denaturing conditions and transferred onto a PVDF membrane. The membranes were blocked with 5% skim milk and then probed with FcRn variants overnight at 4 °C at a 1:100 dilution. The membranes were washed 3 times with PBST, anti-FLAG tag antibody was added at 1:10,000 dilution, and the membranes were further incubated overnight at 4 °C. The membranes were washed with PBST 3 times, probed with anti-goat HRP antibody at 1:10,000 dilution, and incubated for 1 h at room temperature. The membranes were washed with PBST 3 times, developed using ECL Forte, and exposed to an X-ray film.

### 2.8. FcRn and Albumin Binding

Fifty micrograms of each FcRn splice variant was incubated with equal amounts of mouse serum albumin in PBS buffer pH = 5.5 for 4 h at 37 °C. The samples were resolved by SDS-PAGE gel and Western blot.

### 2.9. Mass Spectrometry Analysis

Fifty micrograms of mouse anti NP-OVA IgG was resolved in an SDS-PAGE gel and stained with Coomassie brilliant blue, and the band of interest was cut out from the gel with a sterile blade and placed in a 1.5 mL microfuge tube. The isolated vesicles from the cells infected with FcRn splice variant lentivirus were eluted in a mass spectrometry-compatible buffer. The samples were processed and analyzed at the University of Colorado School of Medicine Biological Mass Spectrometry Proteomics Core Facility.

Electrophoretic protein bands were cut out from Coomassie blue-stained gels, destained, and subjected to in-gel reduction, alkylation, and digested with trypsin overnight as previously described [21]. Subsequently, the samples were acidified with 5% formic acid (FA), and the digested peptides were extracted in 30 µL of 50% acetonitrile–1% FA. The peptides were dried in a vacuum centrifuge, resuspended in 0.1% FA, and subjected to mass spectrometry as described below.

The purified vesicles were digested using a suspension trapping (STrap) protocol as previously described [22]. The various samples were reduced with 5 mM tris(2-carboxyethyl)phosphine), alkylated with 50 mM 2-chloroacetamide, and digested overnight with trypsin (enzyme:substrate ratio 1:50) at 37 °C. In order to recover the peptides from the filter, successive washes with 50 mM TEAB, 0.2% formic acid (FA) and then 50% ACN were used. The fractions subsequently were pooled and dried in a vacuum centrifuge. The recovered digested peptides were aliquoted at 10 μg per sample and eventually were cleaned using Pierce^TM^ C18 Spin Tips (Thermo Scientific) and dried in a vacuum centrifuge. The cleaned samples were resuspended in 0.1% FA in mass spectrometry-grade water.

The digested and cleaned peptides were loaded onto Evotips and analyzed directly using an Evosep One liquid chromatography system (Evosep Biosystems, Odense, Denmark) coupled with a Bruker timsTOF SCP mass spectrometer (Bruker, Bremen, Germany). The purified peptides were separated on a 75 µm i.d. × 15 cm analytical column packed with 1.9 µm C18 beads (Evosep Biosystems, Denmark) and over a 44 min gradient. The buffers used were 0.1% FA in water buffer A and 0.1% FA in acetonitrile buffer B. The instrument control and data acquisition were achieved using Compass Hystar (version 6.0) with the timsTOF SCP operating in parallel accumulation–serial fragmentation (PASEF) mode under the following settings: mass range 1/k/0 start 0.7 V s cm^−2^ end 1.3 V s cm^−2^, 100–1700 *m*/*z*. The capillary voltage was 4500 V, dry gas 8.0 L min^−1^, ramp accumulation times were 166 ms, and dry temp 200 °C. The PASEF settings were: intensity threshold 500, charge range 0–5, active exclusion for 0.2 min, 5 MS/MS scans (total cycle time, 1.03 s), collision-induced dissociation energy 10 eV, and scheduling target intensity 20,000.

The fragmentation spectra were analyzed using the UniProt *Mus musculus* proteome database (proteome ID # UP000000589) by means of the MSFragger-based FragPipe computational platform [23]. Reverse decoys and contaminants were added to the database automatically. The fragment-ion mass tolerance and precursor-ion mass tolerance were set to 20 and 12 ppm, respectively. Fixed modifications were set as carbamidomethyl (C), and variable modifications were set as oxidation (M) and two missed tryptic cleavages were allowed. The protein-level false-discovery rate (FDR) was ≤1%.

### 2.10. FcRn Vesicle Isolation

The method to isolate FcRn vesicles was adapted from [24] with some modifications. The FcRn-knockout podocytes were grown in 10 cm tissue culture dishes. At around 70% confluence, the cells were infected with the FcRn splice variants. After 72 h, the cells were harvested in buffer containing 0.25 M sucrose and 20 mM Tris-HCl, pH 7.0, supplemented with protease inhibitor cocktail, by scraping and then passing through 27 G needles followed by centrifugation at 720× *g* for 5 min at 4 °C to pellet down the nuclei. The supernatants were carefully transferred to new 1.5 mL microfuge tubes and incubated with 10 ug of anti-FLAG tag antibody overnight at 4 °C with shaking. Sheep anti-rat IgG conjugated with Dynabeads (ThermoFisher kit: 11035) was mixed with supernatants for 4 h at room temperature. The beads were collected using a DynaMag™-2 magnet washed with mass spectrometry compatible buffers A 3 times and B 2 times (ThermoFishet kit: 90409). The FLAG-tag labeled vesicles were eluted using the mass spectrometry compatible elution buffer. Similarly, FcRn vesicles were isolated from wild-type podocytes using anti-mouse FcRn antibody and Dynabeads protein G magnetic beads (ThermoFisher kit: 10007D).

### 2.11. Immunofluorescence Staining

Mouse FcRn-KO podocytes were cultured in glass-bottomed MatTek dishes. The cells were infected with FcRn variants 1, 2, 3, 4 and 5 lentiviruses for 72 h. The cells were treated either with 20 μg monomeric IgG or immune complexes (20 μg anti-NP-OVA IgG + 20 μg NP-OVA) for 2 h. Then, the cells were fixed with 4% paraformaldehyde for 15 min. The cells were washed with PBS and permeabilized and blocked (PBS + 0.4% Triton X-100 + 1% BSA + 5% goat serum) for 30 min. The cells were incubated with the primary antibodies overnight at 4 °C. The next day, the cells were washed 3 times with PBS and the secondary antibodies and Hoechst nuclear stain were added for 1 h at room temperature. The cells were washed 3 times with PBS and anti-fade reagent was added to the dishes. The images were acquired using a 100x objective lens, Stedycon Abberior (STED) (Göttingen, Germany), and Olympus IX81 (Tokyo, Japan) confocal microscope system at Colorado University Anschutz Medical Campus Advanced Light Microscopy Core Facility and processed using Huygens Essential 23.10 (Scientific Volume Imaging, Hilversum, The Netherlands) deconvolution software at the conservative setting, and the channels were merged using Fiji ImageJ software (Version: 2.14.0/1.54f).

### 2.12. Statistics

Statistical analysis was performed using Microsoft Excel (Version: 16.77.1) to determine the means and standard deviations.

## 3. Results

### 3.1. Identification of FcRn Alternatively Spliced Variants

Mouse FcRn primers were designed to amplify the full receptor in order to clone the PCR product(s) into a topoTA vector and subsequently subclone it into the final destination lentivirus plasmid (VVPW) to generate a lentivirus to study FcRn overexpression. The PCR reaction yielded the right amplicon size (>1000 bp) using cDNA from different mouse kidneys (Figure 1a). The PCR products were then cloned into topoTA vector. While screening several clones from the topoTA reaction, using colony PCR, several insert sizes were identified (Figure 1b). The positive clones were sequenced, and surprisingly all the clones, regardless of the insert size, were identified as mouse FcRn using the NCBI BLAST tool. The nucleotide sequences of each insert were verified as a potentially valid protein coding sequence (Appendix A) and were traced to their respective exons based on Ensembl.org gene description accession ENSMUSG00000003420 and new variants emerged. All variants contained exons 2 and 7, but expression of exons 3–6 differed among the variants (schematic representation Figure 1c). The four new splice variants were denoted FcRn2, FcRn3, FcRn4, and FcRn5, with FcRn1 representing the full-length receptor. A FLAG-tag was attached at the C-terminus and a lentivirus was produced for each receptor and validated for protein expression by Western blot (Figure 1d). The predicted molecular weight of each splice variant using the Expasy tool was FcRn2 = 30 kDa, FcRn3 = 20 kDa, FcRn4 = 15 kDa, and FcRn5 = 28 kDa. Further protein sequence analysis using Heidelberg’s SMART tool indicated a transmembrane domain located within exon 6, and since this exon is missing from FcRn4 and FcRn5, it suggests that these two variants may be soluble receptors. Western blot analysis showed the presence of FcRn4 and FcRn5 in the supernatants of podocytes overexpressing FcRn4 and FcRn5 (Figure 1e).

We tested the ability of the newly identified splice variants to bind IgG and albumin using far Western blot. Fifty micrograms of mouse IgG and/or albumin were loaded onto native or reduced and denatured SDS-PAGE gels and transferred onto PVDF membranes. The membranes were first probed with 1:100 dilution of the purified recombinant FcRn splice variants and then probed with an anti-FLAG tag goat antibody. Finally, an anti-goat–HRP antibody was applied and the membranes were developed using ECL. Surprisingly, all the splice variants interacted with mouse IgG, but not with albumin at neutral pH. The binding to IgG was abrogated under reduced and denatured conditions. An unknown band around 50 kDa was observed on the membranes of all FcRn splice variants (Figure 1f). As a control, mouse IgG was transferred onto a PVDF membrane and probed with anti-mouse HRP-conjugated antibody to demonstrate the positive presence of mouse IgG. The anti-mouse IgG antibody, however, did not bind to the 50 kDa band. The membrane was then stripped and reprobed with HRP anti-goat antibody which did not interact with mouse IgG (Figure 1g). Taken together, these experiments demonstrate the specificity of the interaction of FcRn splice variants for the recognition of whole mouse IgG and the 50 kDa band observed on the membranes in (Figure 1f). To identify the unknown 50 kDa band, mouse IgG was resolved on an SDS-PAGE gel and the band of interest was cut and analyzed by mass spectrometry. As a result, IgG lambda chain C and V regions and a gamma chain secreted C region were identified (Figure 1h), indicating that FcRn splice variants bind to the IgG Fab fragment.

### 3.2. FcRn1 Function in Monomeric IgG and Immune Complex Trafficking

To examine the role of full-length FcRn (FcRn1) in intracellular trafficking of IgG and immune complexes, mouse FcRn1 was overexpressed in FcRn-KO podocytes using the lentivirus particles. After 72 h of infection, the cells were treated with 20 μg of purified monomeric mouse IgG or immune complexes (20 μg anti-NP-OVA IgG antibody + 20 μg NP-OVA) for 2 h at 37 °C, 5% CO_2_ or left untreated. The cells were washed with PBS, fixed and stained with anti-FLAG tag antibody and antibodies to Rab5 to examine trafficking to early endosomal compartments, Rab7 (late endosomal compartments), Rab11 (recycling compartments), and LAMP1 (lysosomes). Initial observations indicated that FcRn1 was expressed in independent vesicles differently from early, late, recycling, or lysosomal vesicles (Figure 2). Using colocalization threshold analysis from ImageJ, 5 randomly selected fields of view were analyzed and Rcoloc values are reported in Figure 2. Under normal culturing conditions, minimal interaction was observed between FcRn1 vesicles and Rab5 (r = 0.178 ± 0.15), Rab11 (r = 0.188 ± 0.08) and lysosomes (r = 0.265 ± 0.09) but a stronger overlap was observed between FcRn1 and Rab7 vesicles (r = 0.437 ± 0.014). Interestingly, when monomeric mouse IgG was added to the cells, there was an increased interaction between FcRn1 and Rab5 (r = 0.433 ± 0.03) and FcRn1 and Rab11 (r = 0.339 ± 0.05)-containing vesicles. FcRn1 and Rab7 (r = 0.313 ± 0.08) vesicles maintained a noticeable colocalization, but there was no noticeable interaction between FcRn1 vesicles and lysosomes (r = 0.177 ± 0.11). In the presence of immune complexes (ICs, mouse IgG + NP-OVA), a robust interaction remained between FcRn1 and Rab5 (r = 0.482 ± 0.07), 7 (r = 0.334 ± 0.09) and 11 (r = 0.434 ± 0.15) vesicles. There was also a very noticeable colocalization between FcRn1 vesicles and lysosomes (r = 0.439 ± 0.11). In contrast, the FcRn endogenous vesicles appeared to interact mainly with Rab5 vesicles in the presence of ICs (r = 0.332 ± 0.08) and to a lesser extent in the presence of IgG (r = 0.294 ± 0.04) and NT (r = 0.188 ± 0.15). FcRn vesicles interacted minimally with Rab7 vesicles regardless of the treatment NT (r = 0.152 ± 0.05), IgG (r = 0.232 ± 0.03), or ICs (r = 0.174 ± 0.06). Endogenous FcRn vesicles showed a modest colocalization with Rab11 vesicles in the presence of IgG (r = 0.295 ± 0.12) and a minimal interaction in the NT (r = 0.160 ± 0.05) and IC (r = 0.133 ± 0.04) groups. Endogenous FcRn vesicles showed an increased interaction with lysosomes in the presence of ICs (r = 0.295 ± 0.07) and a minimal colocalization was observed in NT (r = 0.182 ± 0.08) and IgG (r = 0.164 ± 0.05) groups (Appendix A).

### 3.3. FcRn Vesicle Characterization and Albumin Binding

We isolated FcRn-specific vesicles from cells overexpressing the FcRn splice variants and from wild-type podocytes using the FLAG-tag or anti-mouse FcRn antibodies. The isolated vesicles were analyzed by mass spectrometry (Figure 3a) (data are available via ProteomeXchange database, accession number PXD045572). Intracellular vesicles are marked with the expression of the Rab-GTPase family of proteins [25]. As such, FcRn-specific vesicles differentially express a large number of Rab proteins as well as the vesicles coating protein clathrin (Appendix A). Also noted was that Rab1A, Rab7A, and Rab14 appear to be enriched in all FcRn variant-specific vesicles. Interestingly, two highly prevalent proteins identified in FcRn vesicles were albumin and immunoglobulin, suggesting that not only do FcRn splice variants recycle the two proteins, but they might also store or sequester them inside FcRn-specific vesicles. To validate FcRn variant interaction with mouse serum albumin (MSA), 50 μg MSA + 50 μg FcRn(s) was mixed in PBS pH = 5.5 for 4 h incubation at 37 °C. The mixture was resolved using an SDS-PAGE gel and Western blot (Figure 3b). The membrane was probed with an anti-FLAG tag antibody and it showed a band shift in the protein sizes for all the splice variants (FcRn(s) + MSA), as the unbound MSA (~70 kDa) appeared as a negative band on the membrane (small arrow). FcRn1, 2 and 5 showed a clear interaction with albumin (large arrows). FcRn3 interaction was not detected. However, in another blot with longer exposure time (Appendix A), it showed a small band of the right size, which may suggest a weak interaction between FcRn3 and albumin at best. As for FcRn4 interaction with albumin, it is unclear whether the binding is possible or not. A supersaturated band that corresponds to the right size is visible above the albumin line. Conversely, FcRn4 might form aggregates at the acidic pH and the band is just an artifact.

The mass spectrometry data suggested an interaction between FcRn vesicles and the nucleus, a feature also observed in the immunofluorescent images (Figure 3c and Appendix A). In addition, many endoplasmic reticulum (ER) and Golgi-specific proteins were identified indicating, the possibility of FcRn vesicles transporting proteins processed in the ER and Golgi apparatus. Mitochondrial proteins were also heavily featured in the mass spectrometry data, suggesting perhaps a constitutive FcRn expression in the mitochondria or FcRn vesicles interacting with the organelle as part of a material exchange mechanism. This interaction was also validated by immunofluorescence using the mitochondrial marker hexokinase 1 and the FLAG-tag antibodies (Figure 3d).

### 3.4. FcRn2 Function in Monomeric IgG and Immune Complex Trafficking

FcRn2 is a splice variant that is missing exon 4. Under normal culturing conditions, FcRn2 vesicles showed a solid interaction with Rab7 vesicles (r = 0.431 ± 0.19) and to a lesser extent with Rab5 (r = 0.329 ± 0.15), Rab11 (r = 0.345 ± 0.11) and lysosomes (r = 0.294 ± 0.19). However, in the presence of mouse monomeric IgG, FcRn2 vesicles showed an increased interaction with Rab11 vesicles (r = 0.420 ± 0.11) and maintained a noticeable colocalization with Rab7 vesicles (r = 0.386 ± 0.11). Some FcRn2/Rab7 vesicles were observed outside the cells. Limited interaction was observed between FcRn2 and Rab5 vesicles (r = 0.265 ± 0.11) and lysosomes (r = 0.248 ± 0.16). By contrast, in the presence of ICs, there was increased colocalization between FcRn2 and Rab5 vesicles (r = 0.455 ± 0.12). FcRn2 vesicle interaction with Rab7 (r = 0.381 ± 0.14) and Rab11 (r = 0.343 ± 0.17) remained noticeable. However, the interaction between FcRn2 vesicles and lysosomes was minimal (r = 0.265 ± 0.09) (Figure 4).

### 3.5. FcRn3 Function in Monomeric IgG and Immune Complex Trafficking

FcRn3 is the splice variant missing exons 3 and 4. As Figure 5 shows, FcRn3 vesicles had a minimal interaction with Rab5 vesicles across the various treatments: NT (r = 0.225 ± 0.11), IgG (r = 0.102 ± 0.06) and ICs (r = 0.069 ± 0.04). There was a noticeable interaction between FcRn3 and Rab7 vesicles in the NT group (r = 0.361 ± 0.13) and in the presence of ICs (r = 0.380 ± 0.18). A limited interaction was observed in the presence of IgG (r = 0.217 ± 0.12). The interaction between FcRn3 and Rab11 vesicles was minimal under NT (r = 0.175 ± 0.10), increased in the presence of IgG (r = 0.392 ± 0.16), and was limited in the presence of ICs (r = 0.262 ± 0.16), with many vesicles observed outside the cells (white arrows). There was very limited colocalization between FcRn3 vesicles and lysosomes: NT (r = 0.297 ± 0.04), IgG (r = 0.212 ± 0.11), and ICs (r = 0.148 ± 0.11).

### 3.6. FcRn4 Function in Monomeric IgG and Immune Complex Trafficking

FcRn4 is the splice variant that is made up of exons, 2, 5, and 7. FcRn4 vesicles do not appear to have a strong interaction with Rab5 vesicles under the various conditions: NT (r = 0.154 ± 0.12), IgG (r = 0.234 ± 0.12), and ICs (r = 0.167 ± 0.02). FcRn4 vesicles showed a modest interaction with Rab7 vesicles under NT (r = 0.309 ± 0.10), a limited colocalization in the presence of IgG (r = 0.254 ± 0.09), and a strong interaction in the presence of ICs (r = 0.58 ± 0.11). FcRn4 vesicle interaction with Rab11 vesicles was limited under normal culturing conditions (r = 0.248 ± 0.08), but increased in the presence of IgG (r = 0.396 ± 0.21) and ICs (r = 0.314 ± 0.26). Lysosomes and FcRn4 vesicles seemed to have a modest interaction in the NT (r = 0.384 ± 0.14) and the IgG (r = 0.411 ± 0.18) groups and a minimal interaction in the presence of ICs (r = 0.229 ± 0.10). Interestingly, it is worth noting that FcRn4 vesicles combined with Rab7, Rab11 and lysosome vesicles were frequently observed to be ejected outside the cells (white arrows), in particular in the presence of IgG or ICs (Figure 6).

### 3.7. FcRn5 Function in Monomeric IgG and Immune Complex Trafficking

FcRn5 is the splice variant missing exons 5 and 6. As Figure 7 shows, FcRn5 vesicles had a minimal interaction with Rab5 vesicles in the presence of normal medium (r = 0.204 ± 0.13); however, there was a noticeable increase in colocalization in cells treated with IgG (r = 0.419 ± 0.19) and ICs (r = 0.555 ± 0.21). There was a minimal association between FcRn5 and Rab7 vesicles in the NT (r = 0.262 ± 0.14) and IgG (r = 0.064 ± 0.06) groups; however, in the presence of ICs, the colocalization increased noticeably (r = 0.450 ± 0.16). It appeared there was a minimal interaction between FcRn5 and Rab11 vesicles in NT (r = 0.197 ± 0.16) and IgG (r = 0.273 ± 0.08) groups, and a much stronger interaction in the IC group (r = 0.554 ± 0.07), with a large number of vesicles released outside the cells. FcRn5 vesicle and lysosome association was minimal in the presence of regular medium NT (r = 0.185 ± 0.09); however, the interaction increased in the IgG (r = 0.409 ± 0.16) and IC (r = 0.360 ± 0.25) groups.

## 4. Discussion

Since mouse podocytes express FcRn at a very low level, we sought to clone the full receptor into a lentivirus plasmid to overexpress the receptor to facilitate the study of its function, particularly in processing monomeric IgG and immune complexes. The cloning procedure yielded several clones of variable insert sizes, but only the clones with insert size ~1000 bp were sequenced, and the clones with smaller inserts were deemed artifacts and discarded. Accidentally, a clone with a smaller insert was sequenced, and to our surprise, that clone was found to be mouse FcRn and a new splice variant. The accidental discovery prompted the sequencing of the other clones with smaller inserts, and three additional mouse FcRn splice variants were identified. Interestingly, all the variants expressed exon 2 and exon 7 where the start and stop codons are located, respectively. However, exons 3, 4, 5 and 6 were variably expressed. This exon configuration renders the amplification of each splice variant individually impossible, since using primers that incorporate the start and stop codons will amplify all splice variants. This is perhaps the reason why FcRn splice variants remained elusive in mouse and human cells until now. Nevertheless, a previous study published in 2008 reported on the identification of one FcRn splice variant in various porcine cells that was primarily localized in the lysosome, as opposed to the endosomal localization of the full-length receptor [26]. The identification of mouse and porcine splice variants suggest that other species may have similar mechanism of cargo sorting involving multiple FcRn splice variants. The clinically relevant question is whether FcRn variants are also present in human cells. Using the same strategy described above to identify mouse FcRn splice variants, we were able to find several human FcRn splice variants (Appendix A), which suggests the complexity of FcRn function in human cells as well.

In order to study the functions of the newly identified mouse splice variants, they were cloned into the lentivirus plasmid VVPW and a virus was generated for each variant. Recombinant FcRn splice variants (including the full receptor) were purified and tested for IgG and albumin binding. Using far Western blot and under native conditions, all FcRn variants bound to mouse IgG, but not to mouse albumin, whereas the interaction between FcRn variants and IgG was abrogated under reduced and denaturing conditions. In addition to IgG binding, an unknown band appeared on the membranes for all the variants. Further mass spectrometry analysis revealed the band is very likely the Fab fragment of IgG. The IgG-binding site is found within exon 4 of FcRn [11,27]. Surprisingly, FcRn 3 and 4 variants lack exons 3 and 4 and still bound to both the IgG Fc portion and to Fab fragments, suggesting the presence of a novel IgG-binding domain within exon 5. Interestingly, mouse FcRn splice variants bound IgG at a neutral pH, though it was previously reported that FcRn disengages from IgG at neutral pH and only binds IgG at acidic pH [28]. However, our data suggest that IgG-FcRn binding/release may not be entirely pH-dependent for mouse FcRn. Indeed, Neuber et al. [29] demonstrated that mouse FcRn bound IgG at both pH 6 and 7.2, whereas human FcRn did not bind IgG at pH 7.2. The interaction between FcRn splice variants and albumin at neutral pH did not yield a positive result. However, the FcRn–albumin interaction was observed at acidic pH most notably for FcRn1, 2, and 5. The interaction of FcRn3 and 4 and albumin is questionable. A weak interaction may exist, but further investigation is required.

Previously, it was reported that FcRn is expressed in vesicles containing the markers Rab5, 7, and 11 or early, late, and recycling endosomes, respectively [12,13]. However, our data indicate that endogenous FcRn or the splice variants are expressed in separate vesicles based on colocalization analysis and mass spectrometry data. FcRn specific vesicles may interact with other vesicles of the endosomal system and the interaction is dependent on the nature of the internalized cargo. For instance, FcRn1 vesicles in the presence of IgG have a very low interaction level with lysosomes, whereas in the presence of ICs, that interaction is increased. At the same time, FcRn1 vesicles interact with Rab5, 7 and 11 vesicles at the same intensity in the presence of IgG or ICs. In contrast, FcRn2 vesicles appear to have very limited interaction with lysosomes in the presence of IgG or ICs. In the presence of IgG, however, FcRn2 vesicles showed a strong interaction with Rab11 and Rab7 vesicles, suggesting that FcRn2 expression may lead to direct IgG recycling. When the cells were treated with ICs, FcRn2 vesicles showed a strong interaction with Rab5 and to a lesser extent Rab7 and 11 vesicles. Taken together, FcRn2 may not be involved in directing cargo to the degradation pathway, but may be a receptor that accelerates IgG recycling. FcRn3 vesicles showed minimal interaction with Rab5 vesicles and lysosomes. However, in the presence of IgG, there was an enhanced interaction, with Rab11 vesicles indicating that the expression of FcRn3 my lead to direct recycling of IgG. However, in the presence of ICs, FcRn3 vesicles interacted with Rab7 vesicles, which led to vesicle ejection outside the cells, perhaps representing a second mechanism by which the podocytes eliminate ICs by ejecting them from the cell instead of degrading ICs in lysosomes. FcRn4 vesicles showed a strong interaction with Rab11 and lysosomes in the presence of IgG. However, in the presence of ICs, FcRn4 vesicles showed a stronger affinity for Rab7 and to a lesser extent Rab11 and lysosomes. One intriguing feature of FcRn4 vesicles is the large number of ejected vesicles outside the cells that demonstrate FcRn4 combining with Rab7, 11, and lysosomes. This finding suggests that FcRn4 expression may lead to IC packaging and release outside the cells, rather than processing them inside the cells. Lastly, FcRn5 vesicles showed a strong interaction with Rab5 and lysosomes in the presence of IgG, indicating that the expression of FcRn5 may lead to IgG degradation. However, in the presence of ICs, FcRn5 vesicles showed a strong interaction with Rab5, 7, 11 and lysosomes, indicating that FcRn5 may function as a scavenger receptor of ICs that are either delivered to lysosomes for degradation or packaged within vesicles and released outside the cells. Our data show that each receptor individually handles the processing of IgG and ICs differently. FcRn-mediated trafficking, however, is likely even more complex than described in our study, as it is likely the interplay between FcRn splice variants and their interaction with other vesicles that determines the fate of the internalized cargo.

To gain more insights into the nature of FcRn vesicles, we isolated FcRn-specific vesicles from all the splice variants and also from cells expressing endogenous FcRn. The isolated vesicles from all the groups were analyzed by mass spectrometry. Surprisingly, albumin and IgG had very high intensity values, strongly suggesting their presence inside FcRn vesicles. This novel finding indicates that FcRn not only binds and recycles IgG and albumin but also that these two important serum proteins can be stored or sequestered inside FcRn-specific vesicles. Given the functional significance of these two proteins, it would be of interest to determine the conditions by which the cells store IgG and albumin and the triggers that cause the release of the two proteins. Clinically, the sequestered IgG may represent a pool of antibodies or harmful autoantibodies that may decrease treatment efficacy or exacerbate the progression of an autoimmune disease.

The mass spectrometry data also suggest the presence of nuclear and mitochondrial materials inside FcRn splice variant vesicles. Immunofluorescence images showed the presence of FcRn variants around the nuclear envelope, as well as surrounding vesicles with nuclear contents. This novel observation extends FcRn function to trafficking of nuclear materials between the nucleus and the cytoplasm.

In addition, FcRn appears to localize within mitochondria, although it is not clear whether FcRn vesicles interact with mitochondria or if FcRn is a constituent of the mitochondrial membrane. To that end, a previous report indicated that FcRn complexed with pemphigus vulgaris IgGs on the surface of keratinocytes and the internalized cargo was trafficked to the mitochondria [30]. This report indicates that FcRn vesicles interact with mitochondria and exchange materials, rather than FcRn being constitutively expressed on the mitochondrial membrane.

## 5. Conclusions

Taken together, our data suggest ever-increasing functions for FcRn. The identification of several alternatively spliced variants in both mouse and human cells indicates the complexity of cargo transcytosis, sorting, and degradation mechanisms that may not be entirely dependent on one receptor. The presence of the newly identified spliced variants and possibly other variants yet to be discovered may help answer some intriguing questions regarding FcRn function, including the transmission of immunity from mother to fetus, the mechanism of IgG and albumin recycling, and the recognition and processing of immune complexes. It may also have a significant impact on FcRn-based treatments such as drug delivery, developing new strategies to suppress autoantibodies in autoimmune diseases, and increasing IgG half-life when this is desired.

## Figures and Tables

**Figure 1 cells-13-00594-f001:**
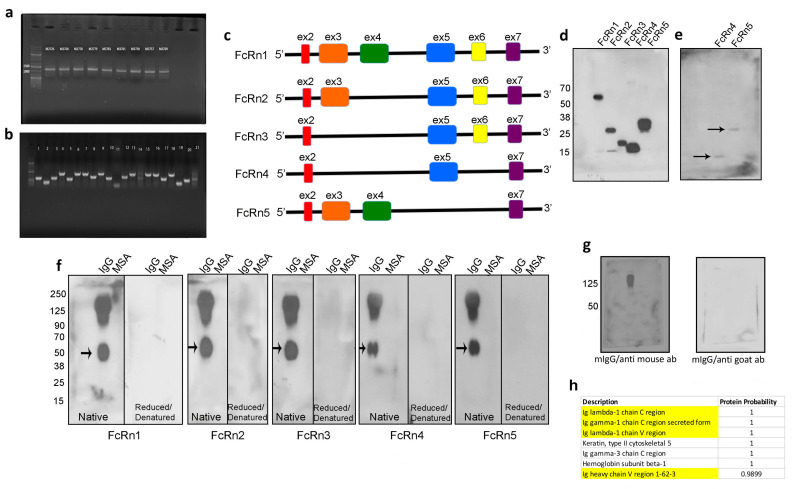
Identification of four mouse FcRn splice variants. Mouse FcRn was amplified by RT-PCR using a primer set that encompasses the full sequence and cDNA prepared from mouse kidneys (**a**) and the PCR products were cloned into a topoTA plasmid. The resulting colonies were screened for a positive insert using colony PCR technique and the cloning primers (**b**). The positive clones were sequenced and the newly identified FcRn splice variants are schematically represented in (**c**). All the splice variants including the full receptor were cloned into a lentivirus vector and produced a recombinant protein that included a FLAG-tag attached to the C-terminus. The cell lysates from infected mouse FcRn-KO podocytes were resolved using SDS-PAGE and Western blot (**d**) and FcRn4 and 5 were also detected in the cell’s supernatants (**e**). FcRn and the splice variants binding to IgG and albumin were determined by far Western blot (**f**). As a control, mouse IgG was resolved on an SDS-PAGE gel, transferred onto a PVDF membrane, and probed with anti-mouse HRP-conjugated antibody, then the membrane was stripped and reprobed with an anti-goat HRP-linked antibody (**g**). The FcRn splice variants recognized an additional unknown band (arrow) in (**f**) that was subjected to mass spectrometry and was identified as IgG light chain of the Fab fragment (**h**) (identified IgG light chain peptides are highlighted).

**Figure 2 cells-13-00594-f002:**
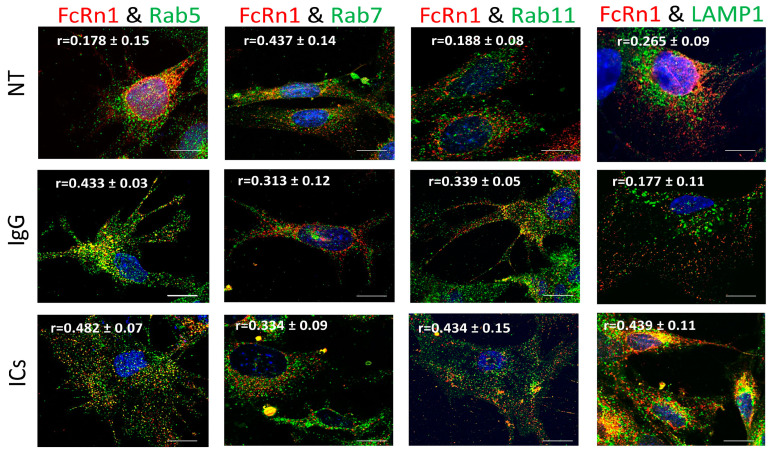
Overexpression of FcRn1 (full receptor) in FcRn-KO podocytes and its interaction with Rab5, 7, 11, and LAMP1 vesicles in the presence of media alone (NT), monomeric IgG or immune complexes (ICs). FcRn-KO podocytes were infected with the lentivirus vector carrying the full FcRn sequence and the cells were assayed after 72 h. Monomeric IgG and immune complexes were added to the cells for 2 h and incubated at 37 °C and 5% CO_2_. The cells were fixed and stained for Rab5, 7, 11, and LAMP1 in addition to anti-FLAG tag to detect FcRn and Hoechst 33342 nuclear stain. The images were obtained using a Stedycon Abberior (STED) microscope and 100x objective lens. The scale bar represents 10 μm. The (r) values represent colocalization intensity plus/minus standard deviation.

**Figure 3 cells-13-00594-f003:**
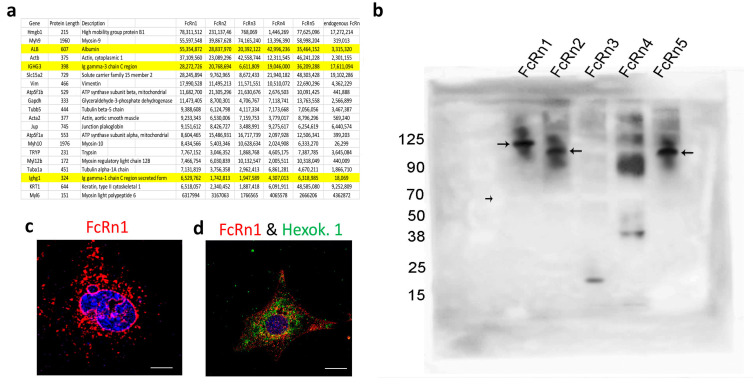
The isolation of FcRn variant specific vesicles and splice variant interaction with albumin. FcRn-specific vesicles were isolated using anti-FLAG tag or anti-mouse FcRn antibodies and protein G conjugated with magnetic beads. The isolated vesicles were analyzed by mass spectrometry, and the 20 proteins with the highest intensity are shown and albumin and IgG are highlighted (**a**). (**b**) FcRn splice variant interaction with mouse albumin was determined by Western blot. Mouse serum albumin (50 μg) was incubated with an equal concentration of purified recombinant FcRn1, 2, 3, 4 or 5 overnight at 37 °C in a PBS buffer at pH 5.5. The solution was resolved using an SDS-PAGE gel and Western blot. The membrane was probed with an anti-FLAG tag antibody and an anti-species HRP-conjugated secondary antibody. Large arrows indicate the interaction of FcRn1, 2 and 5 with albumin. The small arrow indicates the albumin as a negative band. A representative image of FcRn nuclear envelope location (**c**). A representative image of FcRn colocalization with the mitochondrial marker hexokinase 1 (**d**). The scale bar represents 10 μm.

**Figure 4 cells-13-00594-f004:**
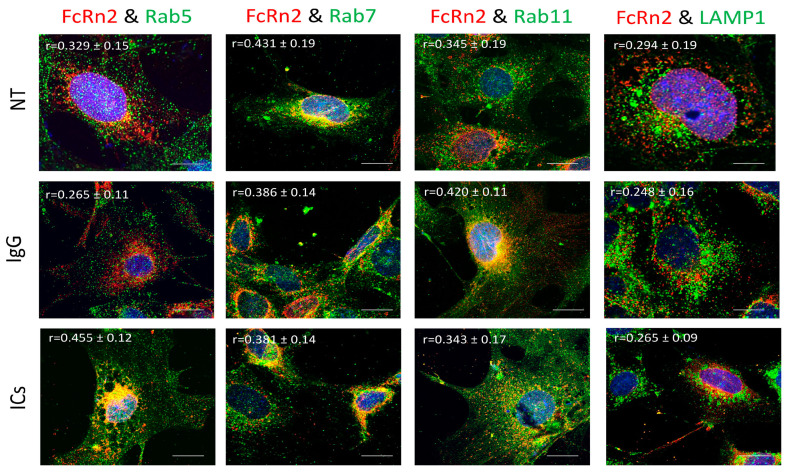
Overexpression of FcRn2 in FcRn-KO podocytes and its interaction with Rab5, 7, 11, and LAMP1 vesicles in the presence of media alone (NT), monomeric IgG, or immune complexes (ICs). FcRn-KO podocytes were infected with the lentivirus vector carrying FcRn2 and the cells were assayed after 72 h. Monomeric IgG and immune complexes were added to the cells for 2 h and incubated at 37 °C and 5% CO_2_. Staining and imaging are as in Figure 2. The scale bar represents 10 μm. The (r) values represent colocalization intensity plus/minus standard deviation.

**Figure 5 cells-13-00594-f005:**
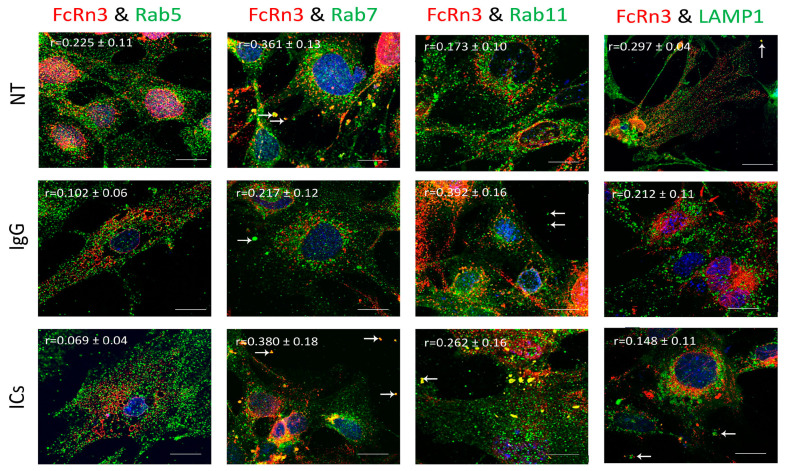
Overexpression of FcRn3 in FcRn-KO podocytes and its interaction with Rab5, 7, and 11 and LAMP1 vesicles in the presence of monomeric IgG or immune complexes. FcRn-KO podocytes were infected with the lentivirus vector carrying FcRn3 sequence and the cells were assayed after 72 h. Monomeric IgG and immune complexes were added to the cells for 2 h and incubated at 37 °C and 5% CO_2_. Staining and imaging are as in Figure 2. White arrows indicate vesicles ejected outside the cells. The scale bar represents 10 μm. The (r) values represent colocalization intensity plus/minus standard deviation.

**Figure 6 cells-13-00594-f006:**
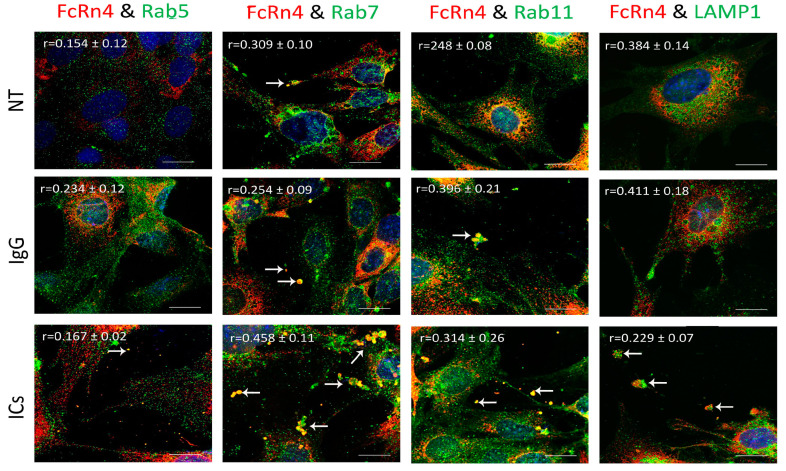
Overexpression of FcRn4 in FcRn-KO podocytes and its interaction with Rab5, 7, and 11 and LAMP1 vesicles in the presence of monomeric IgG or immune complexes. FcRn-KO podocytes were infected with the lentivirus vector carrying FcRn4 and the cells were assayed after 72 h. Monomeric IgG and immune complexes were added to the cells for 2 h and incubated at 37 °C and 5% CO_2_. Staining and imaging are as in Figure 2. The scale bar represents 10 μm. White arrows indicate vesicles ejected outside the cells. The (r) values represent colocalization intensity plus/minus standard deviation.

**Figure 7 cells-13-00594-f007:**
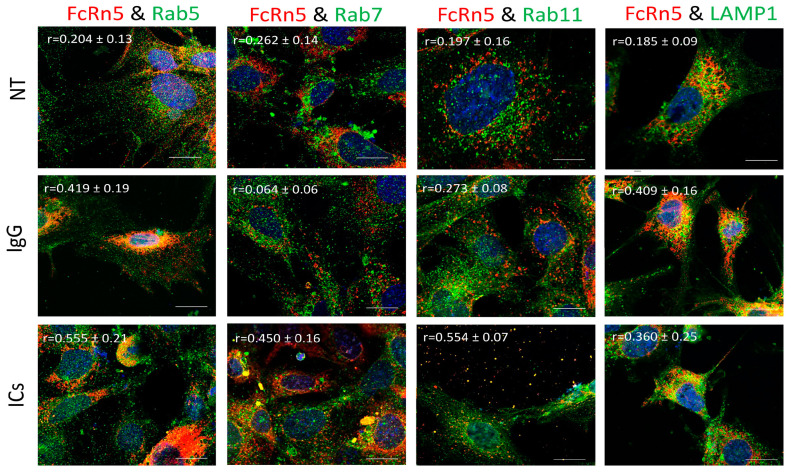
Overexpression of FcRn5 in FcRn-KO podocytes and its interaction with Rab5, 7, and 11 and LAMP1 vesicles in the presence of monomeric IgG or immune complexes. FcRn-KO podocytes were infected with the lentivirus vector carrying FcRn5 and the cells were assayed after 72 h. Monomeric IgG and immune complexes were added to the cells for 2 h and incubated at 37 °C and 5% CO_2_. Staining and imaging are as in Figure 2. The scale bar represents 10 μm. The (r) values represent colocalization intensity plus/minus standard deviation.

## Data Availability

All data generated during this study, including all clones and plasmids, are available for sharing upon request.

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
