# Peer review of "Identification of Four Mouse FcRn Splice Variants and FcRn-Specific Vesicles"

_cells, 2024, doi:10.3390/cells13070594_

Round 1

Reviewer 1 Report

Comments and Suggestions for Authors

The authors identified mouse splice variants and FcRn-specific vesicles. The results are clear and well presented in the figures, tables and supplementary materials. The FcRn variants and their interactions are described using Western blot and far Western blot followed by mass spectrometry analysis. However, we would expect direct interactions of FcRn variants with their ligands with methods capable of assessing affinity. Cell imaging is well performed and contributive. The research process is well presented and leads to original data on FcRn. A few points need clarification/modification:

-          Do the authors have any explanation for the different behaviour of endogenous FcRn and the spliced variant in the colocalisation analysis? Can we completely rule out the involvement of the tag or some other mechanism in this process?

-          Paragraph 3.8: Human FcRn slice variants are also mentioned, although the description is not as well done as for mouse. Human FcRn is encoded by the FCGRT chain, which contains 7 exons and 6 introns. Figure 8 does not seem to integrate these data. Please clarify. The mouse FcRn variants are well described in this paper. Human variants should only be mentioned in the discussion or if integrated in the results, they must be more supported.

-          There is no discussion of the affinity of the spliced variants for their ligands compared to FcRn. Are there any data on this point?

Minor points:

-          Line 37: “Structurally, …functions ” . In fact, structurally, FcRn is related to class I histocompatility complex and belongs to FcgR family

Reviewer 2 Report

Comments and Suggestions for Authors

The part on human FcRn splice variants are incomplete, in that Authors did not investigate the interactions with vesicles.

Authors should recognize that , mammalian FcRn and IgG orthologues have subtle amino acid differences that are the basis for the range of binding affinities observed within and between different species. Notable examples are the lower binding affinity of human FcRn to human IgG3 (due to the presence of R435 instead of H435) and the inability of human FcRn to engage most mouse IgGs (except for weak binding to IgG2b).
